# Modification of Interfacial Interactions in Ceramic-Polymer Nanocomposites by Grafting: Morphology and Properties for Powder Injection Molding and Additive Manufacturing

**Santiago Cano** [1],* , **Ali Gooneie** [2] , **Christian Kukla** [3] , **Gisbert Rieß** [4] , **Clemens Holzer** [1] **and Joamin Gonzalez-Gutierrez** [1],*

1   Polymer Processing, Montanuniversitaet Leoben, 8700 Leoben, Austria; clemens.holzer@unileoben.ac.at
2   Laboratory of Advanced Fibers, Empa–Swiss Federal Laboratories for Materials Science and Technology, CH-9014 St. Gallen, Switzerland; Ali.Gooneie@empa.ch
3   Industrial Liaison Department, Montanuniversitaet Leoben, 8700 Leoben, Austria; Christian.kukla@unileoben.ac.at
4   Chemistry of Polymeric Materials, Montanuniversitaet Leoben, 8700 Leoben, Austria; riess@unileoben.ac.at
*   Correspondence: santiago.cano-cano@unileoben.ac.at (S.C.); joamin.gonzalez-gutierrez@unileoben.ac.at (J.G.-G.); Tel.: +43-3842402 (ext. 3529) (S.C.); +43-3842402 (ext. 3541) (J.G.-G.)

**Abstract:** The adhesion of the polymer to ceramic nanoparticles is a key aspect in the manufacturing of ceramic parts by additive manufacturing and injection molding, due to poor separation results in separation during processing. The purpose of this research is to investigate, by means of molecular dynamics simulations and experimental methods, the role of improved interfacial interactions by acrylic acid grafting-high density polyethylene on the adhesion to zirconia nanoparticles and on the composite properties. The polymer grafting results in high adhesion to the nanoparticles, increases the nanoparticles dispersion and improves the viscoelastic and mechanical properties required for additive manufacturing and injection molding.

**Keywords:** interfacial interactions; rheological properties; mechanical properties; nanocomposites; ceramics; powder injection molding; fused filament fabrication; grafted polymers; molecular dynamics simulation

## 1. Introduction

The composites of thermoplastic polymers with ceramic nanoparticles are used in numerous fields such as medicine [1–3], electronics [4] or flame retardancy applications [3,5]. Moreover, highly filled ceramic-polymer nanocomposites are used for the manufacturing of ceramic parts by processes such as powder injection molding (PIM) [6–9] and fused filament fabrication (FFF) [10–12]. In these processes, the polymer acts as a carrier that enables the shaping of complex parts by injection molding or layer-by-layer extrusion into three dimensional objects. Once the parts are shaped, the polymer is removed in the debinding step. Finally, the parts are sintered to obtain ceramic components with high density.

Polymer nanocomposites are complex systems, whose properties are determined by the properties of the components, the composition, the interfacial interactions and the morphology of the composite [13–15]. The adhesion and wetting of the polymer matrix to the surface of the powder [9,16] is a critical aspect since a poor adhesion will lead to the separation of both materials during processing, resulting in a weak interface and poor macroscopic properties [16,17]. The powder–binder separation is especially

detrimental for the highly-filled composites used in PIM and FFF, since it leads to failure of the process by, for example, filaments with poor mechanical properties which cannot be processed in the FFF machines [11] or by inhomogeneity in the composition of parts with complex geometry in PIM [17,18]. The interaction between the polymers and the ceramic filler is greatly influenced by the polarity of both types of material. In PIM and FFF, multicomponent polymeric blends are used. A major fraction of the polymer blend has low viscosity for PIM, while for FFF not only low viscosity but also flexibility is required. The second major component of the blend, known as the backbone, is in many cases a polyolefin such as polyethylene or polypropylene, which is removed in the last step before sintering and thus must have a high adhesion to the powder. However, polyolefins are non-polar, whereas the surface of the inorganic particles is characterized by a high polarity. Therefore, different strategies have been developed to improve the adhesion and the interfacial interaction between both types of materials.

To improve the adhesion between the polymer and the ceramic, one approach includes the coating of the nanoparticles with a coupling agent such as silane [13] or with a fatty acid such as stearic or oleic acid [6,7,10], which can also be incorporated as a surfactant [8,19]. Fatty acids and stearic acid have been extensively used in PIM and FFF as they improve the dispersion of the particles and reduce the viscosity of the composites [6–8,10,19]. Nonetheless, these low molecular weight polymers are very unstable and degrade if processed at high temperatures or long times [7]. Moreover, short molecules such as the fatty acids do not have strong entanglements with the rest of the polymeric components, which result in poor mechanical properties [13]. Another strategy is the use of polymers grafted with a polar component such as maleic anhydride or acrylic acid, which can be incorporated as a backbone [12,20] or as a compatibilizer [5,21–28]. The improvement of the adhesion to the filler with the use of grafted polymers results in a better dispersion of the filler [27,28] and in an increase in the mechanical properties [5,21–28]. Despite these efforts, a systematic investigation of interfacial interactions is still necessary to understand their effects on the properties of the feedstocks. Here, we aim to address this subject by means of various experiments as well as atomistic insights from molecular dynamics simulations.

In this study, we focus on the effects of interfacial interactions on feedstock properties by grafting of polyethylene with acrylic acid and modifying its adhesion to ceramic zirconia nanoparticles. The resulting composite morphology and properties for the processing by FFF and PIM are addressed. Molecular dynamic simulations are employed to calculate the binding energies between polyethylene with different grafting densities and zirconia. These results are combined with the experimental characterization of composites produced with two commercial grades of high-density polyethylene (HDPE) and yttria-stabilized zirconia powder. One of the HDPE grades is ungrafted, whereas the other is grafted with acrylic acid. The experimental investigation of the adhesion is conducted with the interfacial tension calculated from contact angle measurements and with the infrared spectra of the polymers and filler materials. The results of these experiments are linked to the dispersion of the particles in the nanocomposites and to the viscoelastic and mechanical properties.

## 2. Materials and Methods

### 2.1. Molecular Dynamics Simulation

Computational simulation methods such as atomistic molecular dynamics (MD), Monte Carlo (MC) and dissipative particle dynamics (DPD) are useful tools in the study of the polymer-filler adhesion and the characterization of the interface between both materials [15,29–34]. Moreover, MD simulations have been employed to study the interface between ceramic oxide fillers and polymer chains with grafted maleic anhydride or acrylic acid (AA) [30]. The MD simulations provide detailed atomistic information about the increase in the work of adhesion and the change of the distribution of the polymer at the interface of the composites with grafting [30,33].

MD simulations were employed for the study of the interfacial interactions between zirconia and polyethylene grafted with different contents of acrylic acid: 0, 5 and 30 mol%. Table 1 summarizes the nomenclature used for the different systems in the MD simulations. The bonded and non-bonded interactions were mapped with the optimized atomistic field COMPASS. The simulations were carried out using the Forcite module of the Materials Studio software (BIOVIA Materials Studio 2017 R2, San Diego, CA, USA). Fifty polyethylene chains ($C_{120}H_{242}$) were used in each simulation run. For the polyethylene grafted with acrylic acid, additional side groups were included. The geometries of the polymer chains were initially optimized with energy and force convergence criteria of 0.0001 kcal·mol$^{-1}$ and 0.005 kcal·mol$^{-1}$·Å$^{-1}$, respectively. The polyethylene chains were in contact with a zirconia block with dimensions Lx = Ly ≈ 35.85 Å and Lz ≈ 8.87 Å. An orthogonal simulation cell with dimensions Lx = Ly ≈ 35.85 Å and a xyz Cartesian coordinate system was employed. A vacuum space with a length of 500 Å was put on top of the polymer in the z-direction. Consequently, the total Lz was equal to 635.46 Å. The simulations were run in the NVT ensemble at 443 K for 200 ps with a 0.1 fs time step. The binding energies were calculated and averaged over the final 50 ps of the simulations.

**Table 1.** Materials studied in the molecular dynamics (MD) simulations and nomenclature.

|  | **Polymer** | **Acrylic Acid Content (mol%)** |
|---|---|---|
| PE | PE | 0 |
| 5AAPE | AA-PE | 5 |
| 30AAPE | AA-PE | 30 |

## 2.2. Materials

Tetragonal zirconia ($ZrO_2$) (TZ-3YS-E, partially stabilized with 3 mol% yttria ($Y_2O_3$), Tosoh Europe B.V., Amsterdam, The Netherlands), supplied as spray dried granules with an average particle size of 90 nm and a specific surface area of $7 \pm 2$ m$^2$/g, was the filler used. Figure 1 depicts the scanning electron microscopy (SEM) image of the powder. The commercial acrylic acid-grafted high-density polyethylene SCONA TPPE 2400 (AAHDPE, BYK-Chemie GmbH, Wesel, Germany) employed in our previous study [12] was the functionalized polymer. According to the supplier [35], the material contains a minimum of 5 mol% of acrylic acid and has a melt volume rate (MVR) ranging from 9 to 20 cm$^3$/10 min (190 °C, 2.16 kg). Due to the importance of the viscosity for the processing of polymer nanocomposites, the non-grafted high-density polyethylene was selected based on its MVR. Therefore, the high-density polyethylene CG9620 (HDPE, Borealis AG, Vienna, Austria) was selected as it had a melt flow rate (MFR) of 12 g/10 min (MVR~12.47 cm$^3$/10 min, 190 °C, 2.16 kg). Using the two polymers and the zirconia powder, the compounds shown in Table 2 were compared.

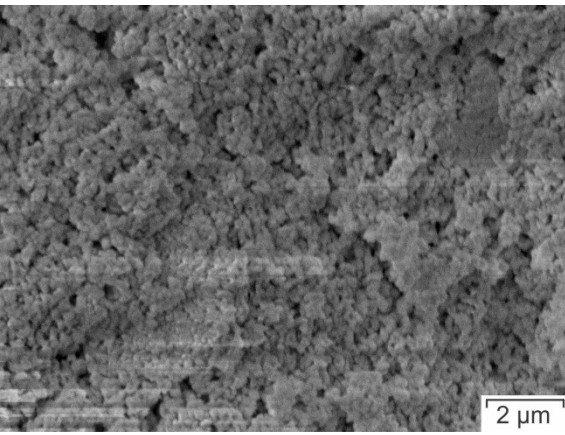

**Figure 1.** SEM of the zirconia powder partially stabilized with yttria.

**Table 2.** Compounds prepared and nomenclature.

|  | Polymer | Acrylic Acid Content (mol%) | Zirconia Content (vol%) |
|---|---|---|---|
| HDPE00 | HDPE | 0 | 0 |
| HDPE30 | HDPE | 0 | 30 |
| AAHDPE00 | AA-HDPE | >5 | 0 |
| AAHDPE30 | AA-HDPE | >5 | 30 |

## 2.3. Preparation of the Compounds

All the materials were processed in an internal mixer fitted with counter rotating roller rotors with a mixing chamber volume of 300 cm$^3$ (HAAKE Rheomix R3000p, Thermo Fisher Scientific Inc., Waltham, MA, USA). Mixing was performed at 160 °C and 60 rpm. The compounding of the nanocomposites started with the filling of the polymer, and after 3 min the powder was added in 5 batches with 5 min after each addition to ensure the stabilization of the torque and thus a proper dispersion. The mixing was continued for a total time of 60 min. The molten compounds were extracted from the chamber and cooled down to room temperature. Following, they were granulated using the cutting mill Retsch SM200 (Retsch GmbH, Haan, Germany) equipped with a sieve with $4 \times 4$ mm$^2$ square shaped perforations. The non-filled systems were processed using the same method in order to avoid the influence of the thermo-mechanical processing step.

## 2.4. Calculation of the Interfacial Tension by Contact Angle Measurements

The contact angle measurements of the unfilled polymers (HDPE00 and AAHDPE00) were conducted using compression molded plates. A plate of each material was produced in the hydraulic vacuum press P200PV (Dr. Collin GmbH, Maitenbeth, Germany). A steel frame with a thickness of 2 mm and flat polytetrafluoroethylene (PTFE) plates were used during the pressing. New PTFE plates were used for each material in order to obtain the same surface roughness, as this is known to influence the contact angle measurements [36]. The compression molding started with the pre-heating of the material at 180 °C during 10 min at a pressure of 1 bar, followed by the compression at 50 bar during 10 min and finally by the cooling down to 30 °C at a pressure of 50 bar.

The contact angle measurements were conducted at room temperature with the goniometer Krüss DSA100 (Krüss GmbH, Hamburg, Germany) using deionized water and diiodomethane as the test liquids. Fifteen repetitions were performed per combination of polymer and liquid. For the zirconia, we used the contact angle values measured by González-Martín et al. [37] for plates of pure zirconia, tetragonal zirconia stabilized with yttria and zirconia with 3% of yttria. The contact angle values were employed for the determination of the surface energy components of each material using the Owens, Wendt, Rabel and Kaelble (OWRK) method [38–40]:

$$\frac{(1 + \cos \theta) \cdot \left( \sigma_l^P + \sigma_l^D \right)}{2 \cdot \sqrt{\sigma_l^D}} = \sqrt{\sigma_m^P} \cdot \sqrt{\frac{\sigma_l^P}{\sigma_l^D}} + \sqrt{\sigma_m^D}, \tag{1}$$

where $\theta$ is the contact angle between the evaluated material (zirconia, AAHDPE or HDPE) and the test liquid; $\sigma_l^P$ and $\sigma_l^D$ are the polar and disperse fractions of the surface energy test liquids, respectively, both known; $\sigma_m^P$ is the polar fraction of the material surface energy, which is calculated as the slope of the fitted line; and $\sigma_p^D$ is the disperse fraction of the material surface energy, calculated as the intercept between the fitted line and the axis. The interfacial tension between the zirconia and each polymer was calculated [41,42] with the equation:

$$\gamma_{zp} = \sigma_z^P + \sigma_z^D + \sigma_p^P + \sigma_p^D - 2 \cdot \left( \sqrt{\sigma_z^D \cdot \sigma_p^D} + \sqrt{\sigma_z^P \cdot \sigma_p^P} \right), \tag{2}$$

in which the subscript $z$ refers to the zirconia and the subscript $p$ to the polymer (HDPE or AAHDPE).

### 2.5. Attenuated Total Reflection Spectroscopy

The infrared absorption spectra of the zirconia powder, the polymers and the compounds were measured by attenuated total reflection spectroscopy (ATR). ATR spectroscopy was performed using a Vertex 70 spectrometer (Bruker, Ettlingen, Germany) at room temperature. A total of 128 scans were accumulated with a resolution of 4 cm$^{-1}$.

### 2.6. Rheological Evaluation

The viscoelastic properties of the neat polymers and the composite materials were evaluated with high-pressure capillary and rotational rheology tests. The high-pressure capillary rheology provides a good understanding of the processability of the materials by FFF or PIM, whereas a more fundamental understanding of the dispersion, structure and interactions between the polymer and filler can be obtained with the rotational rheology tests [43].

The high-pressure capillary rheology was conducted in the capillary rheometer Rheograph 2002 (Göttfert Werkstoff-Prüfmaschinen GmbH, Buchen, Germany) at 160 °C and apparent shear rates from 75 to 2000 s$^{-1}$. Round dies with a diameter of 1 mm and three lengths (10 mm, 20 mm and 30 mm) were employed. Three measurements were conducted with each die and material to ensure the repeatability of the results. The true shear rate and the true shear stress were calculated with the Weissenberg-Rabinowitsch [44] and the Bagley [45] corrections respectively.

Oscillatory rotational rheology tests were conducted in the rotational rheometer MCR 501 (Anton Paar GmbH, Graz, Austria) using parallel plates with a diameter of 25 mm. The tests were conducted using discs with a diameter of 25 mm and 2 mm height; the disc specimens were prepared in a vacuum press with the same procedure as the specimens for contact angle measurements. The oxidation of the material was avoided using an enclosed chamber under a constant nitrogen flow. After the melting of the material, the specimen was compressed to 0.2 mm by taking down the upper plate and the test was started using the automatic control of the normal force to 0 N. The range of evaluated angular frequencies ranged from 0.1 to 500 rad/s using a strain value of 0.1% as the amplitude of oscillation. Three repetitions were conducted per material.

### 2.7. Morphology Analyses

The morphology of the filled materials was studied by scanning electron microscopy (SEM, Tescan Vega II, Tescan Brno, s.r.o., Brno, Czech Republic) of the cryo-fractured surface of the material extruded in the capillary rheometer. The analyses were performed on gold sputtered (100 s at 20 mA) specimens at 5 kV using secondary electrons.

### 2.8. Tensile Tests

The mechanical properties of the materials were measured by means of tensile tests on straight filaments with a diameter of 1.75 mm and a length of 100 mm. This method enables the rapid screening and comparison of similar materials processed under the same conditions [12,46]. The filaments were produced in the Rheograph 2002 at 160 °C using a round die with a 30 mm length and 1.75 mm diameter. The tests were conducted on the universal testing machine Zwick Z001 (Zwick GmbH & Co.KG, Ulm, Germany) with a 1 kN load cell and pneumatic grips. The initial gauge length was set to 50 mm and standardized conditions (23 °C and 50% relative humidity) were used. The tests were conducted at speed of 10 mm·min$^{-1}$ until the rupture of the specimen. Five repetitions were performed per material.

### 2.9. Differential Scanning Calorimetry

The crystallinity of the different polymers and composites was obtained using Differential Scanning Calorimetry (DSC) tests. The measurements were conducted under nitrogen atmosphere on a Mettler Toledo DSC 1 equipped with a gas controller GC 200 (Mettler Toledo GmbH, Vienna,

Austria). Heat-cool-heat runs, with the heating and cooling rate set to 10 K·min−1, respectively, were performed in temperatures between 30 and 190 °C. The normalized enthalpy in cooling was obtained with integral of the crystallization peak for each test. Following this, the crystallinity was calculated as the ratio between the normalized enthalpy in cooling and the normalized enthalpy in cooling for the 100% crystalline polyethylene (293.6 J·g$^{-1}$). Five specimens were investigated per material.

## 3. Results and Discussion

### 3.1. MD Simulations

Figure 2 shows the equilibrium structures of the different polyethylene types on the zirconia surface at the end of the simulation. In all the cases, the polymer chains closer to the zirconia surface are highly oriented parallel to the oxide surface and they get flattened. In the simulated structures of 5AAPE (Figure 2b) and 30AAPE (Figure 2c), the oxygen atoms of the acrylic acid tend to be adsorbed by the zirconia's oxygen atoms and align perpendicular to the surface. The alignment and flattening of the polymer in parallel to metallic and ceramic surfaces has been also observed in other MD simulations [29–31,33]. Considering that the substrate contains oxygen atoms, the oxygen atoms in the polymeric chains localize close to the substrate in order to improve the interactions with the substrate crystalline surface [30]. This phenomenon can be observed in the structures shown in Figure 2 in which the carboxyl groups (C=O) of all the acrylic acid monomers of the chains are in contact with the zirconia surface. Considering that the oxygen plays the most important role in the surface energy of zirconia [37], we evaluated the adhesion of the three types of PE with the zirconia surface by means of the binding energy.

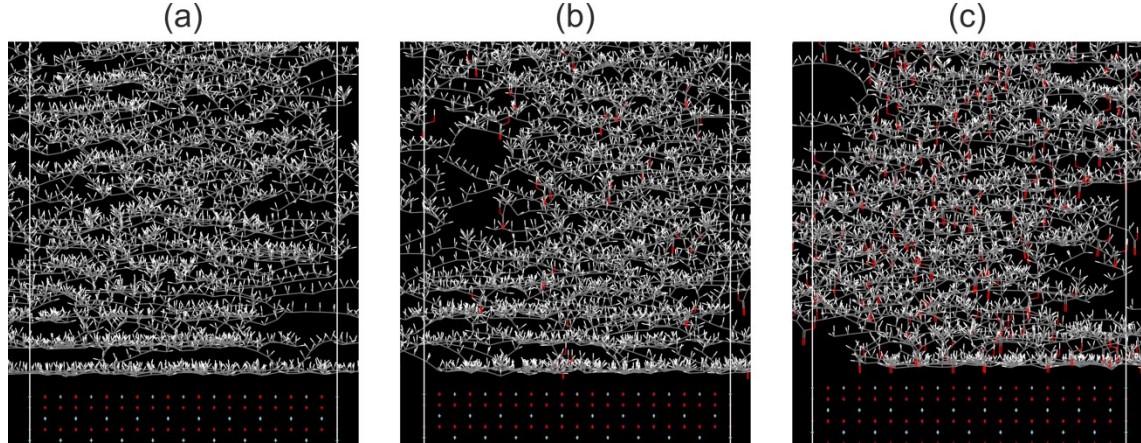

**Figure 2.** The equilibrium structures of (**a**) polyethylene (PE), (**b**) polyethylene grafted with 5 mol% of acrylic acid (5AAPE), and (**c**) polyethylene grafted with 30 mol% of acrylic acid (30AAPE) on top o a zirconia surface. The colors represent the atoms as follows: O: red, C: grey, H: white, Zr: cyan.

The binding energy between two joined materials is directly correlated to the adhesion between them [47]. Table 3 summarizes the binding energy values, with a higher absolute value indicating a better adhesion to the zirconia surface. The binding energy to zirconia is slightly higher for the 5AAPE than for the PE. This difference becomes larger for the 30AAPE, whose binding energy to the zirconia is 78% higher than that of PE. The higher affinity between the polar acrylic acid and the oxide surface and the adsorption of the oxygen atoms of the polymer on the ceramic oxide surface (Figure 2) are responsible of the higher adhesion between polymer and filler [30,33]. Moreover, the binding energy values show that the increase in the grafting level of the polymer can greatly enhance the adhesion between the polymer chains and ceramic nanoparticles.

**Table 3.** Binding energy of the polymers to the zirconia substrate.

| Polymer | Ebinding ($10^5$ kcal·mol$^{-1}$) |
|---------|-----------------------------------|
| PE | −4.2997 |
| 5AAPE | −4.8202 |
| 30AAPE | −7.6500 |

### 3.2. Calculation of the Interfacial Tension by Contact Angle Measurements

Table 4 summarizes the contact angle values measured for compression molded plates of the high-density polyethylene with and without acrylic acid grafting in the virgin and processed state. For the powder, the contact angle values shown in Table 4 are those measured by González-Martín et al. [37] for pure zirconia, tetragonal yttria stabilized zirconia and zirconia with 5 mol% of yttria.

**Table 4.** Contact angle with water ($\theta_w$ and diiodomethane ($\theta_D$) measured for the polymers and values for zirconia as reported by González-Martín et al. [37]. The precision of all the contact angle measurements was within ±2°. Polar ($\sigma^P$) and disperse ($\sigma^D$) components of the surface energy ($\sigma^T$) calculated with Equation (1).

| Material | Contact Angle (°) | | Surface Energy (mN/m) | | |
|----------|------------|------------|------------|------------|------------|
| | $\theta_w$ | $\theta_D$ | $\sigma^D$ | $\sigma^P$ | $\sigma^T$ |
| **AAHDPE00** | 97 | 53 | 32.6 | 0.55 | 33.14 |
| **HDPE00** | 99.3 | 54.2 | 31.9 | 0.34 | 32.24 |
| **ZrO$_2$** | 71.8 | 40.9 | 39.15 | 6.75 | 45.9 |
| **Tetragonal Y$_2$O$_3$ stabilized ZrO$_2$** | 74 | 47 | 35.93 | 6.67 | 42.6 |
| **3%Y$_2$O$_3$-ZrO$_2$** | 66.4 | 40 | 39.61 | 9.14 | 48.75 |

According to the results of Decker et al. [47] and Cao et al. [48] with HDPE surface-grafted with acrylic acid, there is a large decrease in the contact angle of water onto the HDPE, even with low contents of AA in the surface [48]. Despite 5 mol% of AA being used in our study, only a slight difference exists between the values for AAHDPE00 and for HDPE00 (Table 4). Such disagreement might arise from the different distribution of the acrylic acid in the samples here employed [49]. For Decker et al. [47] and Cao et al. [48], all the AA was grafted and concentrated on the surface of the polymer. In our case, the commercial AAHPDE was used to produce the plates employed in the measurements. Therefore, after kneading and pressing, the polar and hydrophilic AA might be in the bulk of the compression molded plates rather than in the surface, and the polar chains might also be oriented towards the bulk and not to the surface.

Using the contact angle values and Equation (1), the surface free energy values reported in Table 4 were obtained. The surface energy and its components were higher for the AAHDPE00 than for the HDPE00. Due to the small differences in the contact angle values, the difference in the $\sigma^P$ of both materials is not as pronounced as expected [47]. Nevertheless, when comparing the values of both components, the $\sigma^P$ of AAHDPE00 is 62% higher than the one of HDPE00, whereas the $\sigma^D$ difference is only of 2%. Thus, it can be stated that the polarity of the AAHDPE00 is higher than the one of HDPE00.

Following this, the interfacial tension between the AAHDPE00 and HDPE00 with the different types of zirconia was calculated with Equation (2) and with the surface energy values from Table 4. The interfacial tension is dependent on the surface energy and the polarity of the materials in contact, and it is inversely proportional to the adhesion between those materials [38,50]. In Table 5, the interfacial tension values between the different material combinations are shown. The interfacial tension of the AAHDPE00 with the three types of zirconia was lower than for the HDPE00. Consequently, a better adhesion to the zirconia can be expected for the AAHDPE than for the HDPE. The increase in the adhesion for the grafted polymer is in agreement with the trend observed in the MD simulation, which showed the increase in the binding energy with the increase in the acrylic acid in the polymer and the

orientation of the acrylic acid towards the oxygen of the pure zirconia surface. The lower interfacial tension for the AAHDPE00 might be also produced by the presence of polar hydroxyl (–OH) groups on the surface of the zirconia plates employed by González-Martín et al. [37]. To determine whether the powder employed in our study contained hydroxyl groups, attenuated total reflection spectroscopy was conducted.

**Table 5.** Calculated interfacial tension between the two types of polymers and different zirconia types.

| | Interfacial Tension (mN/m) | | |
|---|---|---|---|
| **Material** | **$ZrO_2$** | **Tetragonal $Y_2O_3$ Stabilized $ZrO_2$** | **$3\%Y_2O_3$-$ZrO_2$** |
| **AAHDPE00** | 3.74 | 3.46 | 5.53 |
| **HDPE00** | 4.43 | 4.12 | 6.37 |

### 3.3. Attenuated Total Reflection Spectroscopy

Figure 3 shows the infrared spectra obtained for the powder, polymers and composites. The zirconia's peak at 3355 cm$^{-1}$ corresponds to the hydroxyl groups (–OH) bound to the powder surface [7,51–53]. Both unfilled polymers (HDPE00 and AAHDPE00) showed the characteristic polyethylene CH peaks at 1470, 2847 and 2916 cm$^{-1}$ [53,54]. For AAHDPE00 additional peaks were observed at 1167, 1246 and 1700 cm$^{-1}$. The strong peak at 1700 cm$^{-1}$ corresponds to the stretching vibration of the carboxyl (C=O) in the acrylic acid group (–COOH) [48,55].

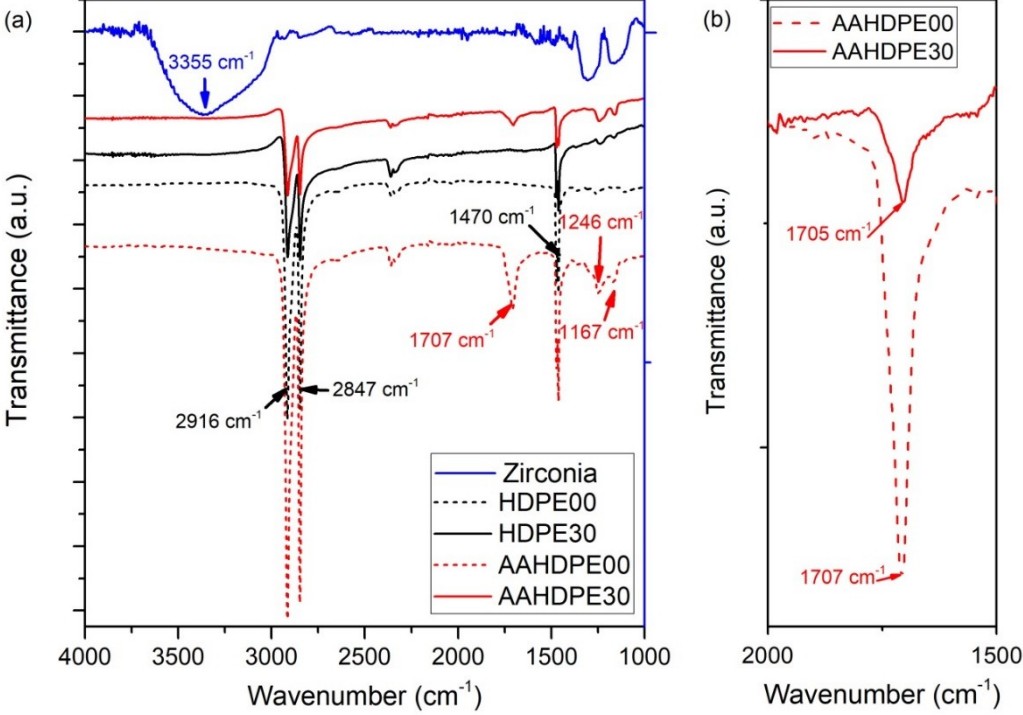

**Figure 3.** (**a**) Infrared absorption spectrum of the HDPE00, HDPE30, AAHDPE00 and AAHDPE30 with the main peaks of the pure components in different colors; (**b**) Magnification of the infrared absorption spectrum for the AAHDPE00 and AAHDPE30 in the 2000–1500 cm$^{-1}$ range with the carboxyl peak for the polymer and the composite.

The reduction in the peaks' intensity for the composites (Figure 3) is produced by the incorporation of the zirconia as a second component. No new peaks appeared due to the chemisorption of the acrylic acid onto the zirconia surface [55] or to the formation of ester links between the carboxyl (C=O) in the acid and the –OH groups in the zirconia [24,56,57]. However, the C=O peak of the AAHDPE has

a slight shift of approximately 2 cm$^{-1}$ after incorporating the powder (Figure 3). The other peaks remained exactly at the same frequencies. The shift of the carboxyl group could be produced by the formation of hydrogen bonds with the hydroxyl groups in the powder oxide surface [58,59] and even with the zirconium dioxide [59] itself. The hydrogen bonding with the oxide would be in agreement with the orientation of the acrylic acid towards the oxygen in the zirconia surface and the high adhesion for the grafted polymers observed in the MD simulations (see Figure 3b,c).

### 3.4. Morphology

The cryo-fracture surface of the two composites, shown in Figure 4, shows the large differences between the morphology of both composites. HDPE30 showed large agglomerates of particles, which are heterogeneously distributed in the polymeric matrix. Contrarily, the zirconia powder is homogeneously distributed in AAHDPE30 without large agglomerates (Figure 4). The differences in the morphologies of both composites can be explained by their different adhesion with the zirconia powder as observed in the binding energy values obtained in the MD simulations at melt temperature (see Section 3.1) and in the interfacial tension values at room temperature (see Section 3.2). The poor adhesion between the non-polar HDPE chains and the zirconia's surface cannot overcome the attractive forces existing between the submicron particles of zirconia, which promote their agglomeration [60]. Furthermore, the hydrogen bonding between the hydroxyl groups in the surface of the powder (see Section 3.3) results in strong powder agglomerates [27,28,61,62]. The use of grafted polymers as compatibilizers [27] has proven to be an effective solution in the reduction in the hydrogen bonding of the hydroxyl groups in silica nanoparticles. Thus, the combination of a high adhesion to the oxide surface and the hydrogen bonding of the acrylic acid with the hydroxyl groups on the powder surface are the responsible factors for the better powder dispersion in AAHDPE30 than in HDPE30.

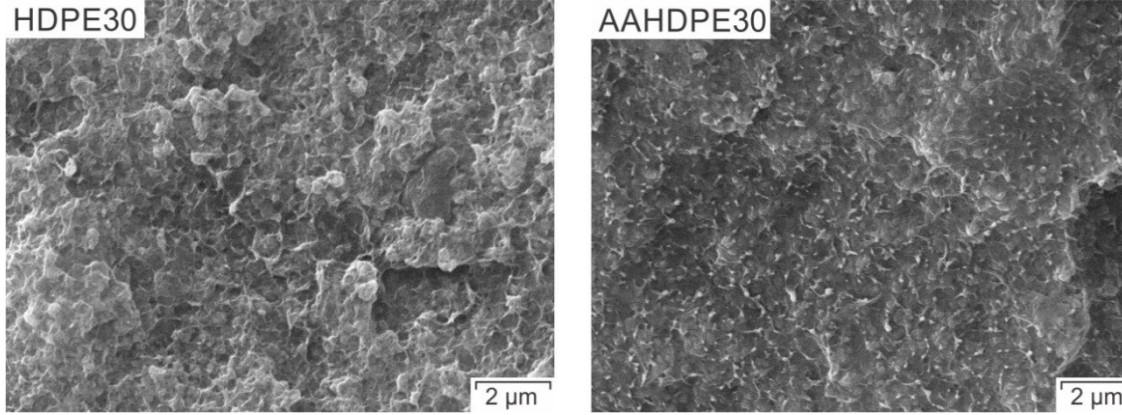

**Figure 4.** Morphology of the strands produced in the capillary rheometer and cryo-fractured of the compounds of HDPE and AAHDPE filled with 30 vol% of zirconia.

### 3.5. Viscoelastic Properties

Figure 5 shows the viscosity measured in a high-pressure capillary rheometer for the different materials. The measurements in the capillary rheometers are strongly related with the processing of the materials by FFF or PIM, since in both processes the material is forced to flow through a narrow nozzle at high shear rates [63]. As can be observed in Figure 5, the shear viscosity of AAHDPE00 is slightly lower than the one of HDPE00 in the range of shear rates evaluated (80 to 1000 s$^{-1}$). On the contrary, the viscosity of AAHDPE30 is higher than that measured for HDPE30, especially at shear rates below 500 s$^{-1}$. Since the effect of the network of particles and polymer plays a more important role at lower shear rates, rotational rheology tests between 0.1 rad s$^{-1}$ and 500 rad s$^{-1}$ were conducted.

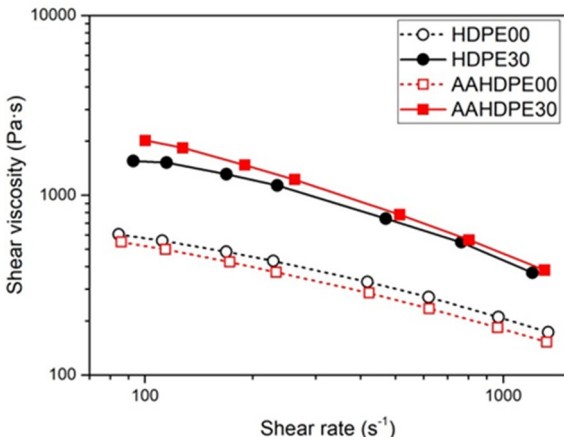

**Figure 5.** Shear viscosity curves as a function of shear rate measured in a round die capillary rheometer at 160 °C for HDPE00, HDPE30, AAHDPE00 and AAHDPE30. Lines are included to facilitate the visualization of the results.

In Figure 6a, it can be observed that the complex viscosity of AAHDPE00 is slightly higher than the one of HDPE00 at low angular frequencies ($<1$ rad·s$^{-1}$), whereas HDPE00 is higher in the rest of the angular frequencies. HDPE00 shows a Newtonian plateau at low angular frequencies, whereas AAHDPE00 exhibits a pseudoplastic behavior in the range of angular frequencies evaluated. The increase in viscosity in the low shear region and the pseudoplastic behavior have been reported for long chain branched polyethylene [64,65] as well as for polyethylene grafted with polar groups such as maleic anhydride, glycidyl methacrylate and acrylic monomers [66,67] or silane [68]. Thus, the differences in the complex viscosity of AAHDPE00 and HDPE might be caused by the branching and partial crosslinking of the poly (acrylic acid) employed in the polymer grafting, which results in a more effective entanglement [67]; the hydrogen bonding between the carboxylic acid in the acrylic acid promotes this behavior [67]. The branching [65] or partial crosslinking [68] of the grafted polyethylene might also be the cause of the trend in the storage modulus (Figure 6b). At low angular frequencies, the G′ of AAHDPE00 is higher than that of HDPE00, whereas the storage modulus of HDPE00 is slightly higher at high angular frequencies. This theory is further supported by the results observed in the loss factor of the unfilled polymers shown in Figure 6c. The slope of the loss factor is negative for HDPE00 in the whole range of angular frequencies evaluated. On the contrary, a positive slope is observed at low frequencies for the AAHDPE00, which is associated with an elastic behavior [68]. Considering the results of the mentioned studies and the trend observed in the complex viscosity (Figure 6), it can be stated that the chain entanglement or crosslinking and the reaction between the acrylic acid monomers in AAHDPE00 are the causes of the big difference in the viscoelastic properties at low angular frequencies for the unfilled polymers.

In the case of the highly filled systems, the complex viscosity of AAHDPE30 is much higher than the one of HDPE30 at low shear rates (Figure 6a). It is known that, at low frequencies, the particle–particle interaction and the network of particles have the main effect on the viscoelastic properties of polymer nanocomposites [43]. As was observed in Figure 4, there are no agglomerates in AAHDPE30, whereas HDPE30 has large agglomerates. Thus, the high viscosity of AAHDPE30 is not produced by the powder's agglomerates. On the contrary, the decrease in agglomerates results in an increase in the particle surface area in contact with the polymer melt, thus increasing the viscosity [43,69–71]. Additionally, the higher adhesion to the powder for the AAHDPE than for the HDPE (observed in the MD simulations at high temperature in Section 3.1) and the hydrogen bonding between the zirconia OH groups and the acrylic acid CO groups (observed in the ATR analyses in Section 3.3) promote the higher viscosity of AAHDPE30 [71–74]. A high work of adhesion results in a thicker layer of polymer absorbed in the particles [72]. At low shear rates, the effective particle

size of the particles is larger due to the entanglement of the absorbed molecules with the rest of the polymer. At high shear rates, the polymer molecules disentangle and orient [72]. The breakup of this network results in a more pronounced shear thinning of AAHDPE30 as compared to HDPE30 in Figure 6 [43]. As can be observed in Figure 6b, the formation of a polymer-filler network due to the improvement of the powder dispersion and the increase in the adhesion also affects the values of the storage modulus [43]. The polymer molecules adsorbed on one particle interact with the chains adsorbed on the near particles and with the free and mobile polymer, resulting in a decrease in the mobility and in an increase in the elasticity for AAHDPE30 as compared to HDPE30 [14,75].

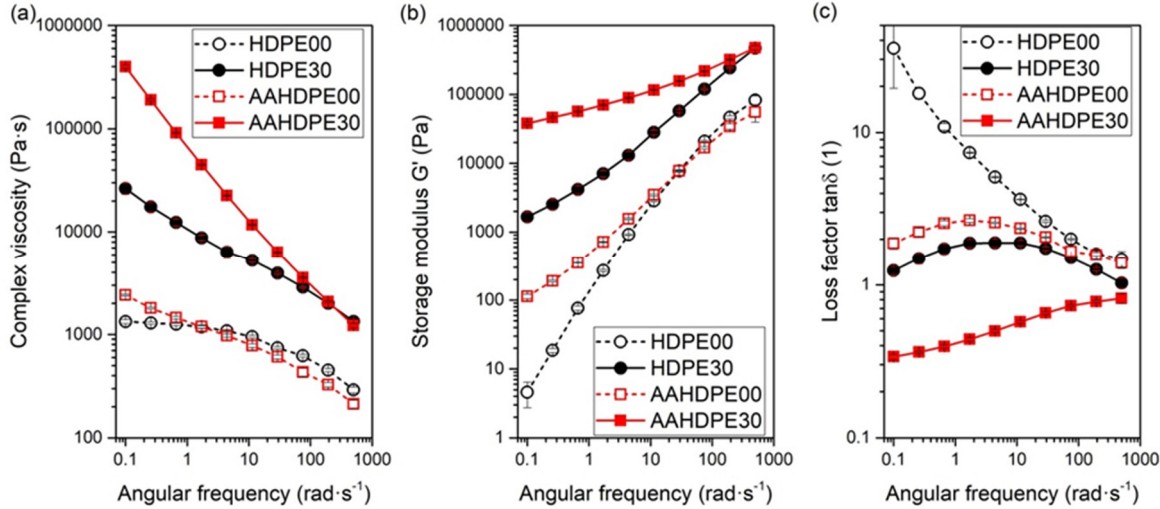

**Figure 6.** Viscoelastic properties as a function of angular frequency measured in parallel plate rotational rheometer at 160 °C for HDPE00, HDPE30, AAHDPE00 and AAHDPE30: (**a**) Complex viscosity, (**b**) Storage modulus and (**c**) Loss factor. Lines are included to facilitate the visualization of the results.

The elastic behavior of the AAHDPE30 can also be observed in its loss factor values, which are smaller than 1 for all the evaluated angular frequencies (Figure 6c). Moreover, the loss factor of AAHDPE30 has a positive slope in all the evaluated shear rates, which corresponds to a pseudo solid behavior. On the contrary, the loss factor of HDPE30 is always higher than 1 and the positive slope only appears at low angular frequencies (Figure 6c). For similar filled polymers, a positive slope and low values of the loss factors can be attributed to a high filler dispersion [76], which is in line with the differences in the morphology of AAHDPE30 and HDPE30.

### 3.6. Tensile Properties

The mechanical properties of the compounds used in FFF are of vital importance [11,12,77]. Sufficient strength and flexibility are required to spool and de-spool the filament for its production and for printing. Moreover, the filaments must be stiff enough to avoid buckling during printing. Furthermore, high strength and stiffness are also required in the PIM process for the de-molding and handling of the parts [78]. Therefore, tensile tests were conducted on filaments produced with all the evaluated compounds. This method enables a rapid screening and comparison of similar materials processed under the same conditions [12,46].

Figure 7 shows representative strain–stress curves of filaments of the materials evaluated, together with the average ultimate tensile strength (*UTS*) and the corresponding average strain value ($\varepsilon_{UTS}$). The secant modulus ($E_S$) was calculated in the strain range between 0.1% and 0.3% in order to avoid the initial stage of the test, in which the slight curvature of the specimens (due to the processing and handling of the filaments) could influence the results. Table 6 summarizes the average and standard deviation values for the three parameters.

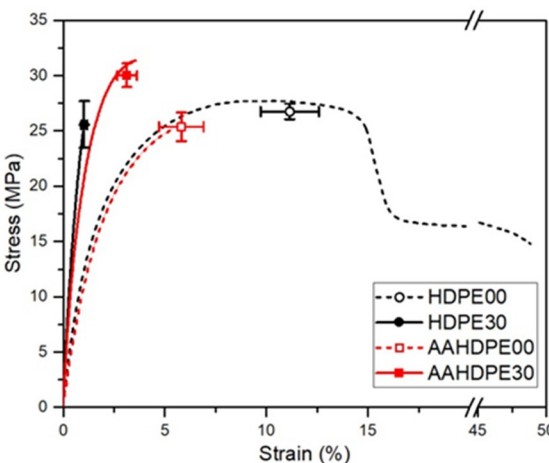

**Figure 7.** Strain–stress curves for HDPE00, HDPE30, AAHDPE00 and AAHDPE30. The average ultimate tensile stress and its corresponding strain are plotted for all the compounds.

**Table 6.** Average and standard deviation of the ultimate tensile strength (UTS), strain at UTS ($\varepsilon_{UTS}$) and secant modulus ($E_S$) values for the HDPE00, HDPE30, AAHDPE00 and AAHDPE30.

|  | HDPE00 | HDPE30 | AAHDPE00 | AAHDPE30 |
|---|---|---|---|---|
| **UTS (MPa)** | 26.75 ± 0.7 | 25.56 ± 2.1 | 25.37 ± 1.3 | 30.01 ± 1.07 |
| **$\varepsilon_{UTS}$ (%)** | 11.15 ± 1.43 | 1.03 ± 0.13 | 5.81 ± 1.1 | 3.13 ± 0.49 |
| **$E_S$ (GPa)** | 1.44 ± 0.05 | 3.41 ± 0.13 | 1.33 ± 0.15 | 2.64 ± 0.09 |

Even though the flowability of the unfilled polyethylene was the only parameter employed to select the two types of high density polyethylene (see Section 2.2), no significant differences existed in the *UTS* and $E_S$ of HDPE00 and AAHDPE00 (Table 6). In fact, the $\varepsilon_{UTS}$ was the only parameter significantly different between HDPE00 and AAHDPE00 (Figure 7) with the non-grafted polymer having a significantly larger value. The incorporation of the particles should result in an increase in the strength and stiffness of the composites [21–23,27,74]. This behavior is observed for AAHDPE30 with an increase of 18% in *UTS* and 98% in $E_S$ with respect to the values of AAHDPE00 (Table 6). In the case of HDPE30, the $E_S$ is 137% higher than the values of HDPE00, whereas no significant difference in the *UTS* was found between HDPE00 and HDPE30 (Table 6). The presence of the zirconia particle results in a decrease in the $\varepsilon_{UTS}$ of 91% in HDPE30 compared to HDPE00 and of 46% in AAHDPE30 compared to AAHDPE00 (Table 6). The decrease in the $\varepsilon_{UTS}$ is produced by the restriction of movement of the polymer by the rigid particles which do not elongate, resulting in a reduction in the ductility of the material [21–23,27,74].

When comparing the properties of the nanocomposites, the $E_S$ of HDPE30 is significantly higher than that of AAHDPE30 (Table 6). The secant modulus, calculated in the strain range from 0.1% to 0.3%, is highly dependent on the polymer structure and factors such as the crystallinity. In order to determine the crystallinity, Differential Scanning Calorimetry (DSC) tests were conducted on five samples of each material. The crystallinity of HDPE00 is 76.21% ± 2.16% and increases to 77.65% ± 0.31% for HDPE30. For AAHDPE00 the crystallinity is 63.42% ± 1.52% and decreases to 57.92% ± 0.85% for AAHDPE30. In polymer nanocomposites with a semicrystalline polymeric matrix, the nanoparticle surface can act as a heterogeneous nucleating site [14]. Smaller nanoparticles are less able to act as nucleating agents than larger fillers [14,79]. Additionally, the improvement of the dispersion of adhesion reduces the mobility of the crystallisable chain segments [68,80]. Therefore, the high dispersion and adhesion to the zirconia results in a lower crystallinity of the matrix in AAHDPE30 than in HDPE30 and consequently in a lower secant modulus [80].

The *UTS* of AAHDPE30 is significantly higher than the *UTS* of HDPE30 (Figure 7). Furthermore, the $\varepsilon_{UTS}$ of AAHDPE30 is three times higher than the one measured for HDPE30 (Table 6). Such

behavior can be attributed to the higher adhesion of the AAHDPE than the HDPE to the zirconia surface, which seems to have a big effect in the high strain region. A high adhesion results in a strong polymer-filler interface and thus in a more effective transfer of load from the polymer matrix to the solid particles [21–23,27,74]. Moreover, the high dispersion of the powder in AAHDPE30 compared to HDPE30 (Figure 4) contributes to difference in the mechanical properties. In polymer composites, defects such as agglomerates or cavities result in a concentration of stresses around those points and eventually the failure of the material in those areas [16,22,23,27]. Thus, it can be concluded that the acrylic acid-grafting of the HDPE results in composites with a higher strength and flexibility by the combination of a strong polymer-filler interface and the reduction in the filler agglomerates.

## 4. Conclusions

The effect of grafting polyethylene with acrylic acid on the interfacial interactions in zirconia nanocomposites for FFF and PIM has been investigated by MD simulations and experimental methods. The MD simulations show the increase in the adhesion to pure zirconia when polyethylene is grafted with acrylic acid, and the orientation of the acrylic acid side groups towards the oxygen in the zirconia. These results have been compared to the experimental characterization of a commercial acrylic acid grafted high density polyethylene and a comparable ungrafted high-density polyethylene and the nanocomposites of both materials with zirconia. The higher polarity of the acrylic acid-grafted high-density polyethylene compared to a similar ungrafted polymer has been observed in their interface with different zirconia surfaces. Moreover, the shift of the carboxyl group peak in the composite with acrylic acid grafted high density polyethylene demonstrates the formation of hydrogen bonds of the acrylic acid with the hydroxyl groups and the oxide in the powder surface.

The increase in adhesion to the powder in the acrylic acid-grafted polyethylene produced by the increase in polarity, the improved interaction with the oxygen in the zirconia surface and the hydrogen bonding to the hydroxyl groups in the powder strongly affect the morphology and properties of the nanocomposites. All these effects result in an improvement in the dispersion of the powder in the polymer matrix in the nanocomposite with the acrylic acid grafted polyethylene. The combination of a higher adhesion between the matrix and filler and better powder dispersion for the acrylic acid grafted polyethylene than for the ungrafted one greatly affects the viscoelastic and mechanical properties of the nanocomposite. The restriction of the polymer chains movement and the formation of a polymer-filler network in the nanocomposite with grafted acrylic acid results in higher viscosity and higher storage modulus than for the ungrafted system, especially at lower shear rates. Furthermore, the reduction of the powder agglomerates that act as defect points and the improvement of the adhesion to the powder provide a higher strength and flexibility for the nanocomposite with grafted acrylic acid than for the one not containing it.

These findings offer insight into the interfacial interactions and the improvement of the adhesion between polymer and filler in ceramic nanocomposites for FFF and PIM by the grafting of the backbone polymer with polar groups. It is shown that these modifications result in an improvement of the quality of the nanocomposite by increasing the filler dispersion and the improving the properties required for its processing. In future studies, the effect of the backbone grafting on the properties and processability of feedstocks for FFF and PIM with multicomponent binder systems will be evaluated.

**Author Contributions:** Conceptualization, S.C., A.G., C.K. and J.G.-G.; Investigation, S.C. and A.G.; Formal Analysis, S.C., A.G. and J.G.-G.; Writing-Original Draft, S.C.; Writing-Review & Editing, S.C., A.G., C.K., G.R., C.H. and J.G.-G.; Visualization, S.C.; Supervision, C.K. and C.H.; Project Administration, C.K. and J.G.-G.; Funding Acquisition, C.K. and J.G.-G. All authors have read and agreed to the published version of the manuscript.

**Funding:** This research was performed under the European project CerAMfacturing and the Austria-China bilateral cooperation projects FlexiFactory3Dp and 3DMultiMat. The CerAMfacturing project has received the financial support of the European Commission in the frame of the FoF Horizon 2020 with financial agreement 678503. FlexiFactory3Dp and 3DMultiMat has received funding from the Austrian Research Promotion Agency under the program Production of the Future, Grant Agreements No. 860385 and No. 875650. Ali Gooneie was supported by the Empa Internal Research Call 2018 in the framework of the "ALLFIN" project.

**Acknowledgments:** The authors greatly appreciate Baris Kaynak, Janak Sapkota and Martin Spoerk for their assistance and the discussions in the planning and testing. The help of Florian Arbeiter in the analysis of the results of the tensile tests is also greatly appreciated. Thanks go to BYK-Chemie GmbH and Borealis AG for kindly donating the polyethylene grades.

**Conflicts of Interest:** The authors declare no conflict of interest. The funders had no role in the design of the study; in the collection, analyses, or interpretation of data; in the writing of the manuscript, or in the decision to publish the results.

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
