# Peer review of "Modification of Interfacial Interactions in Ceramic-Polymer Nanocomposites by Grafting: Morphology and Properties for Powder Injection Molding and Additive Manufacturing"

_applsci, doi:10.3390/app10041471_

Round 1

Reviewer 1 Report

After analyzing this paper, I consider that it is publishable in Applied Sciences journal. However, some corrections should be done:

When tensile properties are discussed (page 12) DSC results are mentioned. Thus, in Materials and Methods section DCS measurements description have to be added. In addition, in Materials and Method section too, the methods should be described in the order of discussion. In Materials and Method section the last method is Morphology test, while in Results and discussion section, the last method is tensile properties. Regarding to ATR results; on one hand, it is impossible to observe anything in the zirconia spectra, the image should be modified to do it more legible. On the other hand, the different peaks mentioned should be indicated in the image, if not it is difficult to understand what it is discussed.

After correcting this minor revision, I consider that the work deserve to be published.

Author Response

Response to Reviewer 1 Comments
The authors thank the reviewer for their positive evaluation of our work. We appreciate the time and effort that the reviewer put in reviewing our manuscript. We believe that the comments have resulted in a significant improvement of the manuscript. Below a detailed response to each comment is given. All amendments and corrections to the original submission are additionally highlighted in yellow color in the uploaded manuscript.

Point 1: When tensile properties are discussed (page 12) DSC results are mentioned. Thus, in Materials and Methods section DCS measurements description have to be added.

Response 1: The details of the DSC measurements have been included in the Materials and Methods section, in the subsection 2.9.

Point 2: In addition, in Materials and Method section too, the methods should be described in the order of discussion. In Materials and Method section the last method is Morphology test, while in Results and discussion section, the last method is tensile properties.

Response 2: Thank you for this suggestion. The Materials and Methods section has been reorganized. Now the morphology analyses are described in section 2.7 and the tensile tests in section 2.8.

Point 3: Regarding to ATR results; on one hand, it is impossible to observe anything in the zirconia spectra, the image should be modified to do it more legible. On the other hand, the different peaks mentioned should be indicated in the image, if not it is difficult to understand what it is discussed.

Response 3: Thank you very much for this comment. Figure 3, showing the ATR, has been redesigned to include the main peaks of the pure components in Figure 3a and the shift in the carboxyl peak for the acrylic acid containing compounds in Figure 3b.

Reviewer 2 Report

This manuscript is good and contains just a few issues:

1) line 37: Power injection molding should not be capitalized.

2) lines 59-61:  This statement is made too early.  Please place it after you discuss prior efforts at improving adhesion between the filler and polymer - i.e., at the end of the following paragraph.

3) lines 74-80: This discussion of modeling interrupts the flow of the Introduction.  Suggest that you move to the Methodology.

4) lines 220-222: No reference to Table 3 and no discussion of the binder energies.

Author Response

Response to Reviewer 2 Comments
The authors thank the reviewer for their positive evaluation of our work. We appreciate the time and effort that the reviewer put in reviewing our manuscript. We believe that the comments have resulted in a significant improvement of the manuscript. Below a detailed response to each comment is given. All amendments and corrections to the original submission are additionally highlighted in yellow color in the uploaded manuscript.

Point 1: line 37: Powder injection molding should not be capitalized.

Response 1: The capital letters have been removed

Point 2: lines 59-61: This statement is made too early. Please place it after you discuss prior efforts at improving adhesion between the filler and polymer - i.e., at the end of the following paragraph.

Response 2: Thank you very much for this comment. We have moved the statement at the end of the next paragraph to improve the clarity of the introduction.

Point 3: lines 74-80: This discussion of modeling interrupts the flow of the Introduction. Suggest that you move to the Methodology.

Response 3: Thank you very much for this suggestion. The paragraph describing the main simulation methods has been moved to section 2.1

Point 4: lines 220-222: No reference to Table 3 and no discussion of the binder energies.

Response 4: We really appreciate this comment. The paragraph discussing the binding energy was lost during the re-formatting of the manuscript to the template of Applied Sciences. We have included it again, and can be found in lines 233-241 of the re-submitted manuscript. Thank you very much.
